# Flight Control Law for Stabilizing Transient Response of the Aircraft during Gun Firing

Chang-ho Ji [1], Chongsup Kim [2],* and Byoung Soo Kim [3]

[1] Flight Control Test Team, Korea Aerospace Industries, Ltd., Sacheon 52529, Republic of Korea
[2] Flight Control Law Team, Korea Aerospace Industries, Ltd., Sacheon 52529, Republic of Korea
[3] School of Aerospace and Software Engineering, Gyeongsang National University, Jinju 52725, Republic of Korea
* Correspondence: robocskim@koreaaero.com

**Abstract:** Highly maneuverable fighter aircraft are equipped with various weapons including a gun firing system for successful air-to-air and air-to-ground missions. In the gun firing system, the muzzle is usually positioned at an offset from the centerline of the aircraft to facilitate maintainability and accessibility on the ground, to ensure the pilot's visibility, and to avoid vibrations. However, this mounting position causes the repulsive force for gun firing to generate a moment around the center of gravity and distorts the aircraft's attitude, degrading the accuracy of the target point. In this paper, we propose the application of an additional augmentation control method, as a hybrid INDI control, that combines model- and sensor-based incremental nonlinear dynamic inversion (INDI) controls to minimize the maximum overshoot of transient response of the aircraft during gun firing. As a result of the frequency- and time-domain evaluation, the additional augmentation control can effectively reduce the transient response during gun firing. In addition, this control method is more robust against uncertainties, and its structure is simple compared to the conventional open-loop type gun compensation control since it does not require any gain scheduling according to flight conditions.

**Keywords:** gun firing; flight control law; attitude stabilization; additional augmentation control; hybrid incremental nonlinear dynamic inversion (INDI)

## 1. Introduction

Most highly maneuverable fighter aircraft are equipped with air-to-air missiles and guns as weapons to gain strategic superiority over other enemy fighter aircraft in air-to-air combat. Air-to-air missiles are weapons used to shoot down target aircraft at a long-distance range. As a representative air-to-air missile, Raytheon's AIM-120 [1], an advanced medium-range air-to-air missile (AMRAAM), is widely used in fighters such as the F-15, F-16, F/A-18, F-22, F-35 JSF, Eurofighter Typhoon, Saab Gripen, and Sea Harrier. In order to reduce the transition response and not degrade handling qualities when the aircraft launches a weapon and the weapon configuration suddenly becomes asymmetric, C. Ji et al. [2] proposed a hybrid incremental nonlinear dynamic inversion (INDI) control, i.e., an additional augmentation control. K. Ahmadi et al. [3] proposed an adaptive modified incremental nonlinear dynamic inversion (MINDI) control to stabilize and control a quad-rotor with partial motor faults. On the other hand, a fighter's gun system is a weapon used at close range in dogfighting. Most fighters, such as the F/A-18E/F Super Hornet [4], F-15K [5], F-22 Raptor [6], and A-50 [7], usually use the M61A2 Vulcan 20 mm gun system. An air-to-air gunnery algorithm [8,9] is utilized to display an aiming symbol on the head-up display (HUD) to improve the accuracy of bullet shooting and provide convenience for the pilot to aim at the target. When air-to-air missile development began actively in the 1960s, it had been argued that a gun system was unnecessary in air combat. Actually, the early F-4 Phantom [10] had no fixed gun, but this argument has subsided after combat experienced

in the Vietnam War where the necessity of a gun system for fighters was shown again. With this lesson, most fighters have adopted a gun system.

The aircraft can be decelerated and have attitude transients when the gun fires due to the gun's lateral and vertical mounting location. The gun's mounting location can also interfere with engine-inlet airflow or pilot vision. The strength and dynamics of local structures and items attached to the gun can cause aircraft vibration upon gun firing [11].

The gun system is usually placed inside the aircraft to maximize the external armament capability and reduce drag. The mounting position of the gun system is selected in consideration of the maintenance, the alignment of the gun control line, the muzzle position, the pilot's view, the recoil force, the damping with the structure supporting the gun system, and the reduction of handling vibration and noise. So, the position of the gun system is determined in the last step after placing the cockpit, engine intake, engine, landing gear, airframe structure, etc. Taking all of these limitations into account, the muzzle of the gun is mounted at an offset from the longitudinal axis of the aircraft.

The open-loop type gun compensation control to reduce the transient response for gun firing is generally used for fighter aircraft, including the T-50/A-50 supersonic advanced trainer/light combat aircraft [12]. For this type of gun compensation control, the control gain needs to be designed depending on flight conditions, so the control gains should be tuned through flight tests. Otherwise, the transient response is greatly affected by the uncertainty of external force caused by gun firing, so many flight test sorties are required.

This paper proposes the application of additional augmentation control method, which is a hybrid INID control based on the angular acceleration measured from the inertial measurement unit (IMU) sensor, to minimize the maximum overshoot of the transient response during gun firing. The main contributions of this paper can be summarized as follows: Firstly, this control method is fairly robust against model uncertainties, effectively reducing the maximum overshoot of transient response and keeping the attitude stable even when the model of the external force exerted on the aircraft during gun firing is uncertain. Secondly, this control method has a simple control structure that does not require a complex control gain scheduling technique since it does not need to be changed according to flight conditions for gun firing. Lastly, above all, the additional augmentation control implemented in the inner loop may dramatically reduce development costs and schedules since many flight test sorties for the control gain scheduling are not required during aircraft development, while control gains of the open-loop type gun compensation control implemented in the outer loop should be tuned through many flight tests during aircraft development since the control gains are designed depending on flight conditions.

The rest of this article is organized as follows: Section 2 introduces the T-50/A-50 gun system and the effect of gun firing as an existing example. Section 3 describes the control method for the open-loop gun compensation, the fundamental NDI control, and the additional augmentation control. Section 4 describes the evaluation flight conditions and shows the evaluation results of the proposed control methods as the frequency-domain analysis and time-domain nonlinear simulation results on the mathematical model of an advanced trainer. Section 5 presents conclusions and future plans.

## 2. Gun Firing Effect

Figure 1 shows the M61A2 Gatling gun system adapted to the A-50 [12], including the gun system layout, and the aircraft transition response characteristics during gun firing.

The 20 mm M61A2 gun was adopted in consideration of the efficiency of the existing logistics support system and the weight limitation. The gun power system applied a hydraulic drive method in which high-speed gun firing can be performed in air-to-air and air-to-ground missions. A port of the gun was designed to quickly disperse noise, vibration, and glare generated during gun firing, to minimize the obstruction of the pilot's visibility, and to block gas from flowing into the engine of the aircraft. A shape of the port was to minimize drag by considering the mounting configuration of the gun while maintaining the aircraft's outer mold line (OML). As a result, the positions of the gun barrel and gun

port were determined to be located offset to the gun reference point on the upper left of the aircraft.

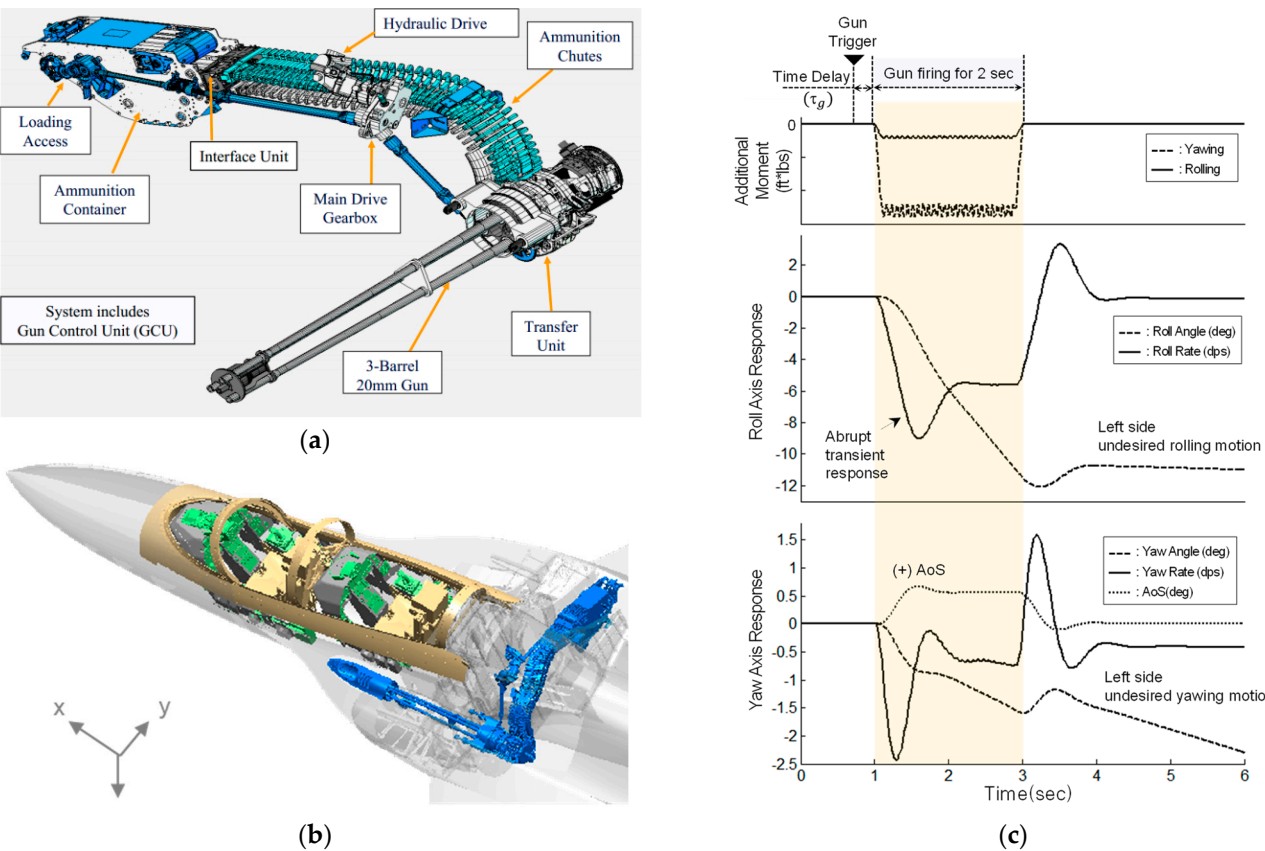

**Figure 1.** A-50 gun system and transient effect during gun firing: (**a**) gun system; (**b**) gun system layout; (**c**) undesired rolling and yawing transient motion during gun firing [12].

During gun firing, the gun firing force is applied at the gun reference point where a pure force acts backward along the gun axis. Since the dominant effect is due to the moments caused by the moment arms from the center of gravity to the gun reference point, the reaction force is additionally applied to all the force and moment directions acting on the center of gravity of the aircraft. The additional moments due to gun firing can be expressed as

$$\Delta L = L_{\delta_{aa}} \Delta\delta_{aa,\,gun} + L_{\delta_{ea}} \Delta\delta_{ea,\,gun} + \Delta L_{gun} \tag{1}$$

$$\Delta M = M_{\delta_e} \Delta\delta_{e,\,gun} + \Delta M_{gun} \tag{2}$$

$$\Delta N = N_{\delta_r} \Delta\delta_{r,\,gun} + \Delta N_{gun} \tag{3}$$

where $\Delta L_{gun}$, $\Delta M_{gun}$, and $\Delta N_{gun}$ are the resultant moments during gun firing, which depend on how and where the gun is mounted on the aircraft and what type of gun it is. $\Delta\delta_{e,\,gun}$, $\Delta\delta_{ea,\,gun}$, $\Delta\delta_{aa,\,gun}$, and $\Delta\delta_{r,gun}$ are resultant control surface deflections of the symmetric horizontal tail, asymmetric horizontal tail, aileron, and rudder, respectively. Here, let $\Delta\delta_{ea,\,gun}$ be $K_{ea}\Delta\delta_{aa,\,gun}$, and then the control surface commands which are additionally required to cancel the resultant moments can be given by

$$\Delta\delta_{e,\,gun} = \frac{-\Delta M_{gun}}{M_{\delta_e}}, \ \ \Delta\delta_{aa,\,gun} = \frac{-\Delta L_{gun}}{L_{\delta_{aa}} + k_{ea}L_{\delta_{ea}}}, \ \ \Delta\delta_{r,\,gun} = \frac{-\Delta N_{gun}}{N_{\delta_r}} \tag{4}$$

where $K_{ea}$ is a value between 0.0 and 1.0, and this value should be scheduled according to the load and control power of the aircraft when maneuvering in each flight condition. In

general, a gun compensation control is applied to compensate for the transient response in the lateral-directional axis, in order to cancel the additional moment generated in the longitudinal axis during gun firing especially to prevent the transient response.

## 3. Flight Control Law Design

This section describes the fundamental INDI control theory, including model-based, sensor-based, and additional augmentation control design concepts, the angular acceleration estimation, the control surface synchronization, the desired dynamics, and the traditional open-loop gun compensation control method.

### 3.1. Fundamental INDI Control Methodology

The INDI control can be classified into model-based, sensor-based, and additional augmentation control design concepts, depending on how it acquires and uses the angular acceleration signals. The model-based INDI, which feeds back the angular acceleration predicted from the aircraft model, was applied to the F-35 joint strike fighter (JSF) [13] and was verified in the entire operational flight envelope. The sensor-based INDI, which feeds back the angular acceleration measured from the IMU sensor, was first applied in the vector thrust aircraft advanced control (VAAC) Harrier [14,15] in 1999. In 2000, NASA applied this control method to an innovative control-effector tailless aircraft [16]. Recently, the Netherlands Aerospace Centre (NLR) and the German Aerospace Center (DLR) in conjunction with the Technical University of Delft in the Netherlands applied this sensor-based INDI to the Cessna 550 demonstrator [17] and proved the performance of the controller in a restricted flight envelope. They have demonstrated the stability and robustness of sensor-based INDI control [18,19]. Lastly, additional augmentation, which combines model- and sensor-based INDI control, is a highly robust control method against model uncertainties. This control method has been used in the F-35 to improve flight performance in the high angle-of-attack and transonic flight range [13]. Recently, C. Kim et al. [20,21] and Jiali et al. [22,23] applied the additional augmentation control method to improve flight performance in high angle-of-attack (AoA) and transonic flight regime.

Figure 2 shows the INDI control structure including all the model-based, sensor-based, and additional augmentation control design concepts. The INDI control structure is separated into a flying quality-dependent portion and an airframe dynamics-dependent portion. In the flying quality-dependent portion, the desired dynamics calculate the desired angular acceleration, $\dot{x}_{des}$, to reflect how the aircraft should fly in response to the pilot control input. The airframe dynamics-dependent portion, which consists of the onboard model (OBM) and control allocation (CA), reflects how the aircraft flies. The OBM provides the current estimated angular acceleration, $\dot{x}_{obm}$, and control effectiveness matrix, $g(x)$. The CA algorithm calculates the change in control surface commands, $\triangle u$, from angular acceleration error, $\triangle d$, to cancel the bare airframe dynamics. Therefore, theoretically, this control method is shown as the flying quality-dependent portion, and the airframe dynamics-dependent portion from the control structure can be isolated if the OBM is accurate.

### 3.2. Model-Based INDI

### 3.2.1. General

The INDI is a control design methodology that cancels the dynamics of the aircraft while simultaneously achieving the desired dynamic response specified by the control law designer [24]. Compared with the existing classical control technology, model-based INDI can effectively improve flight qualities and performance by avoiding complex control gain scheduling and integrating nonlinearities directly into the control law [13,25]. These benefits ultimately result in reduced development costs and time in the aircraft development process.

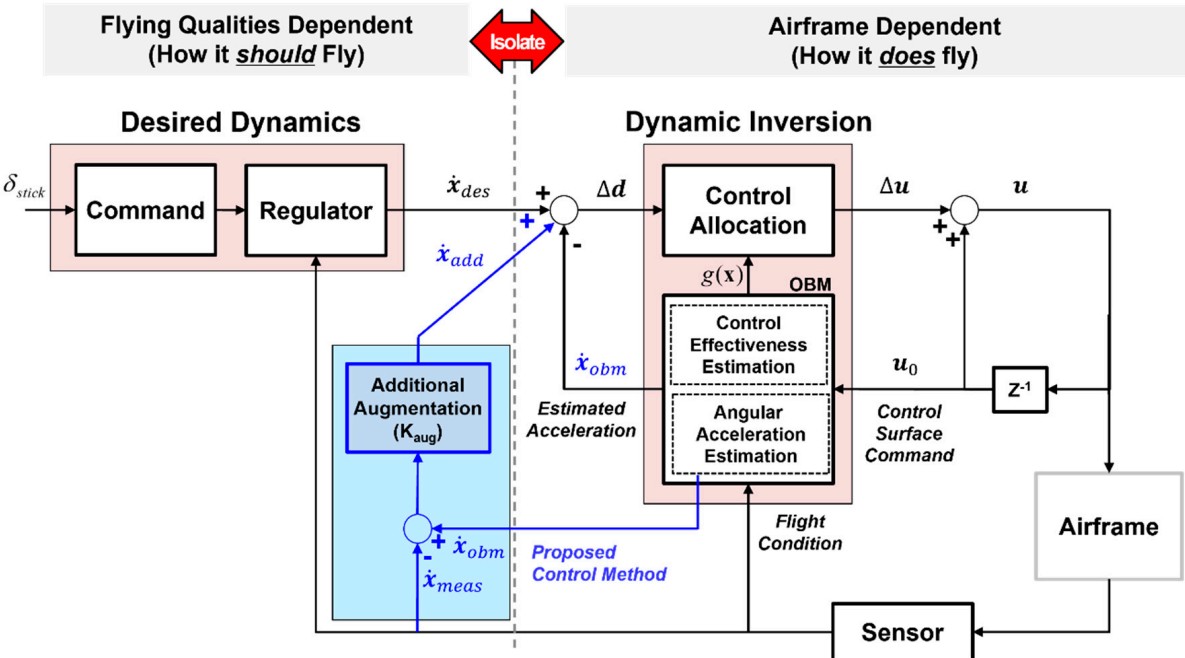

**Figure 2.** Control structure of the INDI control including the model-based, sensor-based, and additional augmentation control design concepts.

For conventional uses where there is a form of small perturbations in trim conditions, the general nonlinear dynamic equation of motion has the form of

$$\dot{x} = f(x) + g(x)u \tag{5}$$

$$y = h(x) \tag{6}$$

where $x \in R^n$ is the state vector, $u \in R^m$ is the control input vector, $n$ is the number of states, and $m$ is the number of control inputs. $f$ represents a nonlinear state dynamic function and $g$ is a nonlinear control distribution function.

The general nonlinear equation, Equation (5), is approximated by Taylor series expansion at the current state $x_0$ and control input $u_0$ as

$$\dot{x} = f(x_0) + g(x_0)u_0 + \frac{\partial}{\partial x}[f(x) + g(x)u]_{u_0, x_0}(x - x_0) + \frac{\partial}{\partial u}[f(x) + g(x)u]_{u_0, x_0}(u - u_0) \tag{7}$$

The higher-order terms above second derivatives are not included because these terms are assumed to be negligible [22,23,26,27]. The summation of the first and second terms is expressed as $\dot{x}_0$. The assumption of $x = x_0$ considers that controls change much faster than the states based on the assumptions of high computational time and instantaneous control effectors, meaning that the change of state between two steps is negligible as a result of time scale separation between states and control inputs. The increment in control input is defined as $\Delta u = u - u_0$. Then, Equation (7) can be rewritten as

$$\dot{x} \approx \dot{x}_0 + g(x_0)\Delta u \tag{8}$$

We will specify $\dot{x}$ as the rate of the desired control inputs, $\dot{x}_{des}$, to achieve the flying quality requirements. By swapping $\dot{x}$ in the previous equation to $\dot{x}_{des}$, Equation (8) can be arranged into Equation (9).

$$\Delta u = g(x_0)^{-1}(\dot{x}_{des} - \dot{x}_0) \tag{9}$$

Consequently, the new control command, $u$, can be given by combining the previous control inputs, the control matrix, and the desired control inputs as

$$u = u_0 + g(x_0)^{-1}(\dot{x}_{des} - \dot{x}_0) \tag{10}$$

The formulation of model-based INDI control is obtained by simply replacing $\dot{x}_0$ in Equation (10) with modeled dynamics $f_{obm}(x) + g_{obm}(x)u_0$ as

$$u = u_0 + g_{obm}(x_0)^{-1}[\dot{x}_{des} - (f_{obm}(x_0) + g_{obm}(x_0)u_0)] \tag{11}$$

In the case of $g_{obm}(x_0) \approx g(x_0)$ and $f_{obm}(x_0) \approx f(x_0)$, we can remove the incremental control $u_0$, and Equation (11) can be expressed as

$$u = g(x_0)^{-1}(\dot{x}_{des} - f(x_0)) \tag{12}$$

It can be seen that the simplified version of model-based INDI is identical to the NDI control method. The performance of this control method depends on model accuracy [18]; that is, if an accurate aircraft model can be obtained through wind tunnel testing, the desired dynamics can be achieved according to the flight quality requirements set by the control law designer without considering the aircraft dynamics. However, there are uncertainties in the mathematical dynamic models of the aircraft, which are generally obtained through wind tunnel tests. In particular, it is quite difficult to obtain an accurate dynamical model in the atmospheric state of the high angle-of-attack flight region where there is unpredicted flow dynamics [20,21]. In addition, it is difficult to obtain an accurate mathematical model since the control system has high-order characteristics and due to the computational time delay of the control system and several model characteristics such as actuator and sensor dynamics.

### 3.2.2. Control Surface Command Synchronization

The control surface command feedback loop for INDI control plays a vital role in the dynamic inversion loop since it is assumed that the angular acceleration signal and the control surface command feedback signal are available at the same moment in time. This assumption implies that phase lags and time delays affecting the signal path should be carefully considered, and the phases of signals are matched where possible. Especially, for the sensor-based and additional augmentation control INDI, it is mandatory that the control surface command feedback signal be synchronized adequately with the filtered angular acceleration signal. To achieve high-order synchronization matching, second- or fourth-order synchronization filters need to be in the control surface feedback path to eliminate the latency of the angular acceleration feedback signal [28].

$$H_{syn} = \frac{\omega_{syn}^2}{s^2 + 2\zeta_{syn}\omega_{syn}s + \omega_{syn}^2} \ or \ \left(\frac{\omega_{syn}^2}{s^2 + 2\zeta_{syn}\omega_{syn}s + \omega_{syn}^2}\right)\left(\frac{\omega_{syn}^2}{s^2 + 2\zeta_{syn}\omega_{syn}s + \omega_{syn}^2}\right) \tag{13}$$

At this time, it is very important to select where the control surface command feedback is obtained from because the effect of the feedback signal on the aircraft structural vibration below 10 Hz frequency band is different according to the position to which the control surface command signal is fed back. With these considerations, this paper applies a control surface feedback method using a fourth-order synchronization filter which C. Kim et al. proposed in order to eliminate structural vibration caused by the noise elements in the control surface feedback signal [24].

### 3.2.3. Desired Dynamics

The INDI design method maintains a close connection with desired dynamics which can be formed differently such as proportional dynamics [29], proportional plus integral dynamics [11], flying quality dynamics [30], and ride quality dynamics [31] in order to directly map flying quality parameters into the control law design. So, it has the advantage of applying the existing traditional flying quality specifications such as MIL-STD-1797A [11] by designing readily desired dynamics with classical control theories. The desired dynamics consist of a command shaping and a regulator. The command shaping aims to translate the pilot stick input to the desired aircraft behavior, and the regulator aims to directly set the low-order equivalent system (LOES) parameter values of the control system, which are the short-period mode damping and natural frequency in the longitudinal axis, the roll mode time constant, Dutch roll mode damping, and natural frequency in lateral-directional axis, to comply with the traditional flying quality specification while the aircraft is performing missions.

Figure 3 represents the control structure of desired dynamics. Figure 3a shows the desired dynamics architecture in the longitudinal axis [21]. As a longitudinal axis response type, the normal acceleration, Nz, is selected in the focus of gross acquisition to achieve fast response during air-to-air combat maneuvers in up and away (UA) configuration. The control structure of the desired dynamics in the longitudinal axis is a proportional-plus-integral (PI) type with feedback variables of Nz and pitch rate, $q$. In addition, the feed-forward control gain and the pilot prefilter on the pilot command loop are designed to improve the initial pitch angular acceleration and optimize the handling qualities. The control gains in the desired dynamics are scheduled with Mach number and altitude to ensure a satisfactory level of flying qualities in the entire flight envelope. The initial values of the flying quality parameters in the longitudinal axis can be obtained as

$$K_{ni} = \frac{g_0}{V_T}\omega^2, \ K_q = 2\zeta\omega, \ K_{np} = \frac{g_0}{V_T}T_{\theta2}\omega^2, \ K_f = \frac{g_0}{V_T}T_{\theta2}\omega^2 \tag{14}$$

$$\frac{T_{\theta2}^{des}s + 1}{T_{\theta2}s + 1} = \frac{K_{fn}s + 1}{K_{fd}s + 1} \tag{15}$$

where $\zeta$ and $\omega$ are the damping ratio and natural frequency of the short-period mode, $V_T$ is the aircraft's true speed (ft/s), $g_0$ is the gravitational acceleration (g), i.e., 1 g = 9.81 m/s², and $T_{\theta2}$ is the pitch attitude time constant, and these values are obtained from the aircraft dynamics.

Figure 3b,c show the desired dynamics architecture in the lateral-directional axis. The stability-axis roll rate response type is selected to achieve fast roll response in the lateral axis, and the stability-axis sideslip response type is selected to augment Dutch roll damping and frequency in the directional axis. So, the desired dynamics can be designed on a basis of proportional control with feedback variables of stability-axis roll rate $p_s(°/s)$, sideslip $\beta(°)$, and sideslip rate sideslip $\dot{\beta}(°/s)$.

The initial values of flying quality parameters in the lateral axis and directional axis can be obtained as

$$K_{r1} = K_{r2}\frac{p_{s,\,max}}{p_{s,cmd,max}}, \quad K_{r2} = -\tau_{roll}, \tag{16}$$

$$K_{y1} = K_{y2}\frac{\beta_{max}}{\beta_{cmd,max}}, \quad K_{y2} = \omega_{dr}, \quad K_{y3} = -2\zeta_{dr}\omega_{dr} \tag{17}$$

where $p_{s,max}$ and $p_{s,cmd,max}$ are the maximum roll rate and maximum roll rate command, $\beta_{max}$ and $\beta_{cmd,max}$ are the maximum sideslip and maximum sideslip command, $\tau_{roll}$ is the roll mode time constant, and $\zeta_{dr}$ and $\omega_{dr}$ are the Dutch roll mode damping and natural frequency which are design goals of lateral directional control.

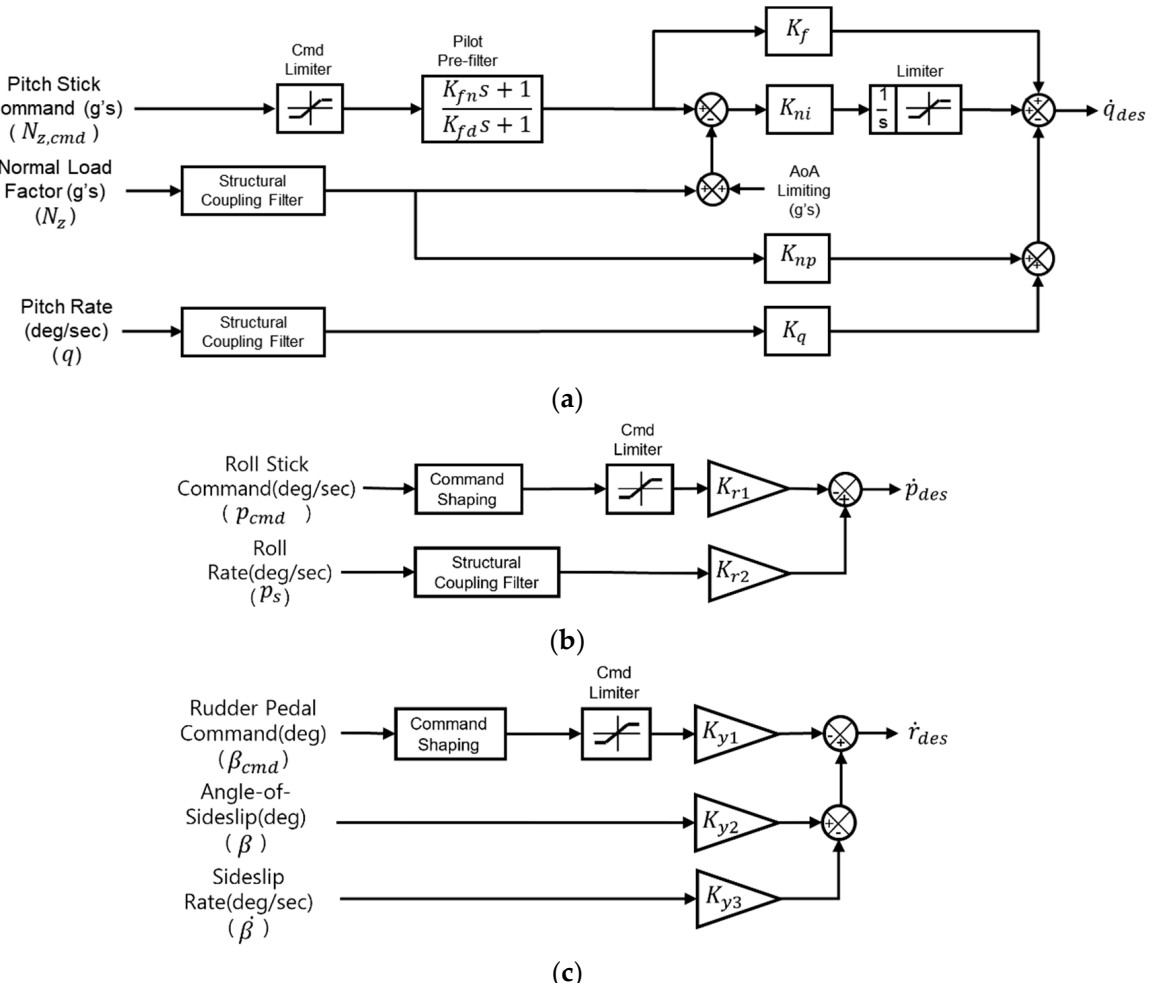

**Figure 3.** Control structure of desired dynamics: (**a**) longitudinal axis; (**b**) lateral axis; (**c**) directional axis [20,21].

### 3.2.4. Gun Compensation Control

Gun compensation control is generally applied to fighter aircraft in order to prevent the transient response during gun firing. It is also used in the desired dynamics of the model-based INDI control. Figure 4 shows the control law structure of the traditional open-loop type gun compensation control to compensate for the objectionable yawing and rolling moment due to the gun firing reaction force. The gun compensation control generates the control commands of additional asymmetric horizontal tails, ailerons, and rudder control surface commands as

$$\triangle \delta_{aa,gun} = K_{g3} \triangle \delta_{gun}, \ \triangle \delta_{r,gun} = K_{g4} \triangle \delta_{gun}, \ \triangle \delta_{ea,gun} = K_{ea} \triangle \delta_{aa,gun} \tag{18}$$

where $\triangle \delta_{gun} = sig_{GT} \frac{a\,\tau_g}{s+a} \left\{ K_{g2}K_{g5} - K_{g1}(1 - K_{g2}) \right\}$.

The $sig_{GT}$ represents the gun burst signal which changes from 0.0 to 1.0, and the gun trigger note is the switch signal to engage the gun compensation control when all the conditions of the note are true. $\tau_g$ is the time lag between gun burst signal activation and actual gun firing. The control gains, $K_{gx}$, in the control loop are scheduled with Mach number, altitude, and airspeed to reduce the transient motion at the time of firing the gun according to each flight condition.

For this open-loop type gun compensation control, the control gains should be tuned through many flight tests during aircraft development since the control gains are designed

depending on flight conditions so that the transient response is little affected by the uncertainty of external force caused by gun firing.

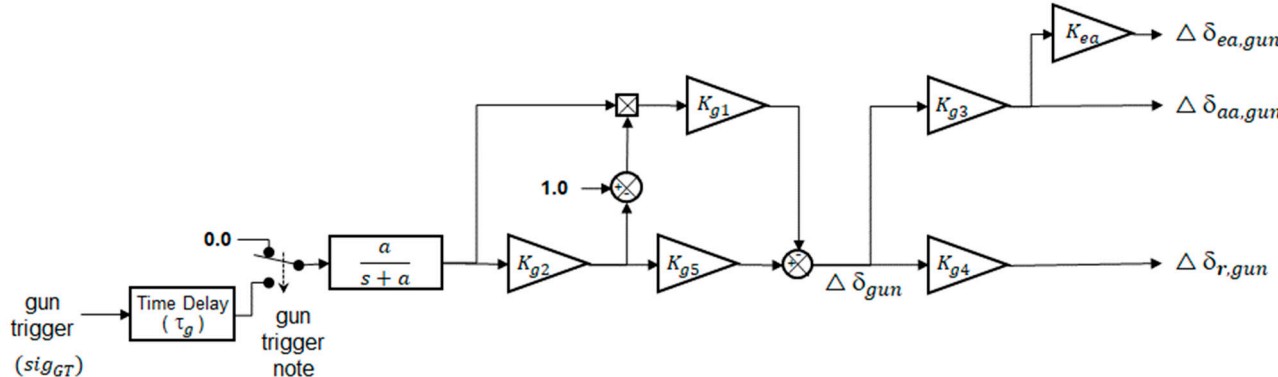

**Figure 4.** Structure of open-loop gun compensation control.

### 3.3. Sensor-Based INDI

From Equation (10), the new control command, $u$, combining the previous control inputs, the control matrix, and the desired control inputs is shown as

$$u = u_0 + g(x_0)^{-1}(\dot{x}_{des} - \dot{x}_0) \tag{19}$$

Equation (19) shows the sensor-based INDI control that uses the angular acceleration, $\dot{x}_0$, measured directly from the IMU sensor, and this control method is considerably robust in response to system model uncertainties as the system dynamics, $f(x)$, does not need to be included. This control structure is simple so that it does not require a complex control gain schedule and gun compensation control; i.e., it does not need many flight conditions to tune the control gains for gun firing.

The sensor-based INDI control [17] that feeds back the measured angular acceleration from the IMU sensor has been proposed as a way to solve the model-based INDI's weakness and the dependence on the model accuracy. This control method is a variant of model-based INDI that reduces the dependence of the control laws on the aircraft model while maintaining the advantages of model-based INDI so that it can improve the control law robustness in response to model uncertainties. However, it is fairly difficult to measure an accurate and reliable angular acceleration due to time lag, bias, and noise characteristics in the real world. Since the time delay characteristic of measured angular acceleration has the disadvantage of reducing the phase margin of the control system [32], there is a limitation in applying it to the entire flight envelope of a production fighter aircraft that satisfies the airworthiness certification criteria [21].

### 3.4. Angular Acceleration Estimation

The INDI control feeds back the angular acceleration state, which is the embodiment of the force and moments of the aircraft. The INDI control can remove the effect of degrading flying qualities due to various complex factors, and the system performance can be improved. There are two ways to estimate the angular acceleration of the aircraft. One is to estimate it from the theoretical model; the other is to estimate it using sensor information. The former method has little system time delay, but it is model-dependent, so the control performance can be significantly affected by system model uncertainties in the actual system. However, the latter method does not depend on the system model parameters, so it can effectively rule out the effects of external disturbances as well as system model uncertainties.

Until now, it has been difficult to obtain an angular acceleration signal with high accuracy because the sensors for directly measuring the angular acceleration signals could not have been used in the aircraft due to technical issues [26,27,33]. For this reason, the angular acceleration signals have been obtained through signal estimation methods [34,35]. The most practical estimation method usually used is to differentiate the angular velocity from the inertial measurement unit sensor, but there is a disadvantage in that the differential operation itself also causes significant amplification of high-frequency noise of the control system, resulting in very low signal accuracy [36]. Therefore, it is necessary to design a low-pass filter with limited bandwidth to eliminate the high-frequency noise which causes a large phase lag. However, the design to suppress the amplified noise and minimize phase lag at the same time is quite difficult to implement [37]. To overcome this shortcoming, approaches to estimate angular acceleration based on recursive linear smoothed Newton (RLSN) [38], Kalman filter [39], and complementary filter [40] theories have been proposed, but their performance has not yet been proven in aircraft production. In this paper, we consider the method of differentiating the angular velocity from the inertial measurement unit sensor to obtain the angular acceleration. To eliminate the noise from both the measured angular rate differentiation and the high-frequency structural coupling effect, the second-order SCF is designed on the feedback path of angular acceleration with the synchronization filter at the control surface command feedback path. The angular estimation system in the presence of uncertainties and sensor noise and the effect of the sensor noise on the outputs are discussed in our previous papers [22,41]

### 3.5. Additional Augmention Control

As an alternative to this disadvantage of the model- and sensor-based INDI controls, this paper proposes the application of additional augmentation control, a hybrid INDI control, which is based on the angular acceleration measured from the inertial measurement unit (IMU) sensor. This section describes an additional augmentation control that can minimize the maximum overshoot of the transient response during gun firing. The control surface synchronization used for the control is also discussed.

The additional augmentation control [20–23] can compensate for the disadvantages of model- and sensor-based INDI control by reflecting the characteristics of the aircraft in the operational flight envelope. In the control structure of Figure 2, the control command can be expressed as

$$\boldsymbol{u} = \boldsymbol{u_0} + \boldsymbol{g}_{obm}(\boldsymbol{x})^{-1} \left[ \dot{\boldsymbol{x}}_{des} - \left\{ \boldsymbol{K_{aug}} \dot{\boldsymbol{x}}_{meas} + \left( \boldsymbol{I} - \boldsymbol{K_{aug}} \right) \dot{\boldsymbol{x}}_{obm} \right\} \right] \tag{20}$$

where

$$\dot{\boldsymbol{x}}_{obm} = \boldsymbol{f}_{obm}(\boldsymbol{x}) + \boldsymbol{g}_{obm}(\boldsymbol{x}) \boldsymbol{u_0}$$

$$\dot{\boldsymbol{x}}_{add} = \boldsymbol{K_{aug}} \left( \dot{\boldsymbol{x}}_{meas} - \dot{\boldsymbol{x}}_{obm} \right)$$

$$\Delta \boldsymbol{d} = \dot{\boldsymbol{x}}_{des} - \dot{\boldsymbol{x}}_{obm} - \dot{\boldsymbol{x}}_{add}$$

where $\dot{\boldsymbol{x}}_{obm}$ is estimated angular acceleration from OBM, $\dot{\boldsymbol{x}}_{add}$ is additional angular acceleration based on the error which is a mix of the measured and estimated angular acceleration, $\Delta \boldsymbol{d}$ is the virtual control command, and $\boldsymbol{K_{aug}}$ is an n-dimensional diagonal matrix which means the control gain set as a ratio of $\dot{\boldsymbol{x}}_{obm}$ and $\dot{\boldsymbol{x}}_{meas}$. The detailed structure of the additional augmentation control was discussed in our previous paper [41]. Each element of $k_i$ has an arbitrary value between 0.0 and 1.0. By substituting Equation (20) into Equation (5), the dynamic equation of motion including the control law is bounded as expressed in

$$\dot{\boldsymbol{x}} = \boldsymbol{f}(\boldsymbol{x}) + \boldsymbol{g}(\boldsymbol{x}) \boldsymbol{u_0} + \left[ \dot{\boldsymbol{x}}_{des} - \left\{ \boldsymbol{K_{aug}} \dot{\boldsymbol{x}}_{meas} + \left( \boldsymbol{I} - \boldsymbol{K_{aug}} \right) \dot{\boldsymbol{x}}_{obm} \right\} \right] \tag{21}$$

For any $1 \leq i \leq n$ and $1 \leq j \leq$ m,

$$min\{e_{i,meas}, e_{i,obm}\} \leq f_i(x) + \sum_{j=1}^{m} g_{ij}(x)u_0 - \left[\{k_i \dot{x}_{i,meas} + (1 - k_i)\dot{x}_{i,obm}\}\right] \leq max\{e_{i,meas}, e_{i,obm}\} \qquad (22)$$

where

$$e_{i,meas} = f_i(x) + \sum_{j=1}^{m} g_{ij}(x)u_0 - \dot{x}_{i,meas} \qquad (23)$$

$$e_{i,obm} = f_i(x) + \sum_{j=1}^{m} g_{ij}(x)u_0 - \dot{x}_{i,obm} \qquad (24)$$

where $f_i(x)$ is the $i$-th element of $f(x)$, $g_{ij}(x)$ is the $(i,j)$-th element of $g(x)$, and $k_i \in \{x \mid x \in \mathbb{R}, 0 \leq x \leq 1\}$ is the $(i,j)$-th element of $k$. In addition, $\dot{x}_{i,meas}$ and $\dot{x}_{i,obm}$ are the $i$-th elements of $\dot{x}_{meas}$ and $\dot{x}_{obm}$, respectively.

Since plant dynamics cannot be accurately modeled in the real world, it is difficult to accurately replace the unique plant dynamics with the desired dynamics. In general, the additional augmentation control is a control synthesis technique that cancels out the dynamic properties of a dynamic system and replaces them with the desired dynamics chosen by the control law designer. Especially, in the case of a highly maneuverable fighter aircraft with a wide operational flight range, it is difficult to precisely model the dynamic model in the transonic speed or high angle-of-attack flight regime, where the flow field is very unsteady. In the existing model-based INDI in that flight regime, a significant number of flight tests should be carried out to improve the model's accuracy. As a result, the increased number of flight tests increases the development cost and program period of the aircraft, increasing the price of aircraft production and thus weakening price competitiveness.

The additional augmentation control INDI control proposed in this paper can use the selected angular acceleration according to the flight conditions, which mixes the estimated angular acceleration and the measured angular acceleration. That is, the use proportion of $\dot{x}_{obm}$ increases in the subsonic and supersonic flight regions where the relatively accurate model can be estimated, and the use proportion of $\dot{x}_{meas}$ increases in the high angle-of-attack and transonic flight regimes, where it is difficult to estimate the model accurately. Therefore, this control method prevents the flight qualities from being deteriorated by applying the bounded error of the maximum angular acceleration. It can not only ensure the robustness of the system, but also improve the dynamic response characteristics of the system. In addition, the additional augmentation INDI control uses the angular acceleration measured from the inertial measurement unit (IMU) sensor in the transonic regime. This control structure is simple, so it does not require a complex control gain schedule. So, the additional augmentation control does not need many flight conditions to tune the control gains for gun firing since it does not apply the open-loop type gun compensation control in the transonic regime using the sensor-based INDI.

## 4. Analysis and Evaluation Result

### 4.1. Evaluation Points and Method

As representative flight conditions to evaluate transient response during gun firing, Mach numbers 0.4 and 0.8 at altitude 10 kft and Mach number 0.8 at altitude 40 kft were selected. At the flight conditions, the frequency-domain linear analysis was performed in order to evaluate the basic flying qualities on the lateral-direction axis. As LOES criteria, the Dutch roll mode damping and frequency, the roll mode time constant and spiral stability, the Gibson phase rate, and the bandwidth are selected. The time-domain simulation is also performed to compare the transient responses between the open-loop type gun compensation control and the proposed additional augmentation control. In the time-domain simulation, the control system robustness is analyzed by applying the uncertainties of reaction force and the time delay during gun firing in the case that the transient response

characteristics are not reflected in the control gain scheduling when the gun compensation control is designed.

Table 1 shows the results for designing control gains of the desired dynamics and the gun compensation control in each flight condition. For the fighter aircraft operating in a wide flight envelope, the control gains of desired dynamics are optimized in each flight condition in order to obtain the desired flying qualities in the entire operational flight envelope, when $K_{aug}$ between 0.6 and 1.0 is applied. The control gains in the gun compensation control are designed in the flight envelope in order to reduce the aircraft transient response to the reaction force at the time of gun firing. However, the modeling of the aircraft transition response to the reaction force is inevitably inaccurate since it is difficult to measure the accurate reaction force caused by firing the gun in a wind tunnel test. So, the control gains are finalized by tuning in the flight test phase.

**Table 1.** Control gains of desired dynamics and gun compensation.

| Mach | Airspeed (knots) | Alt (kft) | $K_{aug}$ | Inner-Loop Control Gains | | | | | Gun Compensation Control Gain | | | | | |
|---|---|---|---|---|---|---|---|---|---|---|---|---|---|---|
| | | | | $K_{r1}$ | $K_{r2}$ | $K_{y1}$ | $K_{y2}$ | $K_{y3}$ | $K_{g1}$ | $K_{g2}$ | $K_{g3}$ | $K_{g4}$ | $K_{g5}$ | $K_{ea}$ |
| 0.4 | 220 | 10 | | −2.9 | −2.9 | −12.2 | 12.3 | −4.9 | 1.4 | 0.2 | 0.3 | 0.5 | 0.6 | 0.6 |
| 0.8 | 448 | 10 | 0.0 | −4.6 | −4.6 | −22.9 | 22.9 | −6.7 | 1.4 | 0.003 | 0.2 | 0.5 | 0.6 | 0.6 |
| 0.8 | 346 | 30 | | −2.9 | −2.9 | −16.7 | 16.7 | −5.7 | 2.8 | 0.04 | 0.1 | 0.4 | 0.6 | 0.6 |
| 0.4 | 220 | 10 | | −2.9 | −2.9 | −12.2 | 12.3 | −4.9 | N/A | N/A | N/A | N/A | N/A | N/A |
| 0.8 | 448 | 10 | 0.6~1.0 | −4.6 | −4.6 | −22.9 | 22.9 | −6.7 | N/A | N/A | N/A | N/A | N/A | N/A |
| 0.8 | 346 | 30 | | −2.9 | −2.9 | −16.7 | 16.7 | −5.7 | N/A | N/A | N/A | N/A | N/A | N/A |

*4.2. Modeling of Additional Moments Due to Reaction Force*

As already mentioned in Section 2, the gun barrel and gun port of the A-50 fighter are mounted at an offset to the upper left on the aircraft's principal axes. Due to this offset mounting, the reaction force during gun firing generates a yaw moment and a rolling moment in the negative (−) direction at the center of gravity. A pitching moment is also generated, but it is not considered in this paper since it is very small and neglectable. Therefore, the only additional rolling moment, $\Delta L_{gun}$, and yawing moment, $\Delta N_{gun}$, caused by the gun reaction force are modeled based on the experimental data, which were previously obtained during gun firing, as shown in Figure 5.

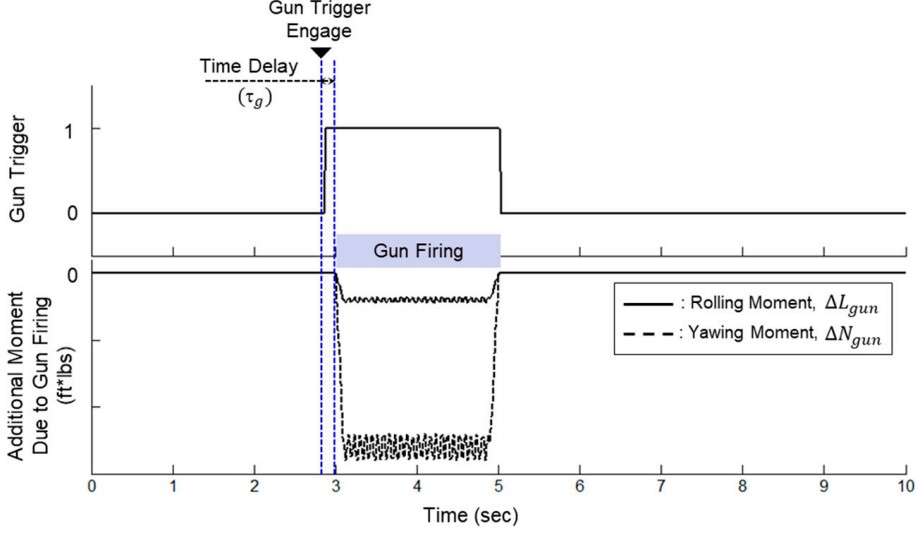

**Figure 5.** Modeling of additional moment due to reaction force during gun firing.

The gun compensation control is designed in consideration of the amount of time delay which is a time interval between the pilot's gun activation time and the actual gun firing starting time. The compensation control is finally verified through flight tests.

### 4.3. Flying Quality Analysis

The lateral-directional flying quality specifications can be divided into two categories as the first-tier criteria and the second-tier criteria. The first-tier criteria are requirements that should be satisfied, which are the Dutch roll mode damping and frequency, the roll mode time constant, and the spiral stability. These criteria are directly applied to the control gain optimization process [21] in the desired dynamics. The second-tier criteria are guidelines that are not involved in the control gain design process but are used as references for designers to judge flying qualities, which are the bandwidth and the Gibson standards. The levels of flight quality are classified into 1, 2, and 3. Most aircraft aim to achieve level 1 flying qualities as a design goal in MIL-STD-1797A [11].

Table 2 shows the frequency-domain linear analysis results of the Dutch roll mode damping and frequency, the roll mode time constant, and the spiral mode root using the low-order equivalent system (LOES) analysis method which is used to satisfy these requirements in the design and its evaluation process since it is difficult to conduct a direct evaluate the control system, which normally a high-order system (HOS), including the aircraft model, sensors, actuators, and control laws.

**Table 2.** Result of equivalent system analysis for each control method and $K_{aug}$.

| Mach | Alt (kft) | Control Method | $K_{aug}$ | Dutch Roll Mode Freq. (rad/s) | Damping | Roll Time Const. (sec) | Spiral Root (sec$^{-1}$) | Mismatch Cost | HQ Level |
|---|---|---|---|---|---|---|---|---|---|
| 0.4 | 10 | Gun Comp. | - | 3.91 | 0.65 | 0.42 | N/A | 4 | 1 |
| | | Additional Augmentation Control | 0.6 | 3.99 | 0.66 | 0.41 | N/A | 14 | 1 |
| | | | 0.8 | 4.03 | 0.66 | 0.41 | N/A | 20 | 1 |
| | | | 1.0 | 4.09 | 0.66 | 0.41 | N/A | 26 | 1 |
| 0.8 | | Gun Comp. | - | 6.09 | 0.58 | 0.28 | N/A | 4 | 1 |
| | | Additional Augmentation Control | 0.6 | 6.08 | 0.64 | 0.29 | N/A | 15 | 1 |
| | | | 0.8 | 6.10 | 0.65 | 0.29 | N/A | 20 | 1 |
| | | | 1.0 | 6.17 | 0.66 | 0.29 | N/A | 25 | 1 |
| 0.9 | 30 | Gun Comp. | - | 4.95 | 0.70 | 0.39 | N/A | 5 | 1 |
| | | Additional Augmentation Control | 0.6 | 5.00 | 0.70 | 0.41 | N/A | 18 | 1 |
| | | | 0.8 | 5.04 | 0.70 | 0.42 | N/A | 23 | 1 |
| | | | 1.0 | 5.09 | 0.70 | 0.42 | N/A | 30 | 1 |

As the value of $K_{aug}$ increases, the value of the mismatch cost function increases; the value of the mismatch cost function is a standard for judging the reliability of the LOES analysis, and its value of 10 or less is recommended to guarantee the reliability of the LOES analysis in MIL-STD-1797 [11]. Figure 6 shows the gain and phase frequency responses of HOS and LOES of the control system including the computational time delay and actuator and sensor dynamics, with $K_{aug} = 1.0$ at Mach number 0.8, altitude 30 kft. The two frequency responses within 20 rad/s, which is a frequency band that affects flight qualities, are considered to have similarity by comparison; that is, the LOES analysis result seems to be still reliable, even though the value of the mismatch cost function is 10 or more. The Dutch roll mode frequency and the damping ratio, over 1.0 and 0.35, respectively, tend to increase as $K_{aug}$ increases. The Dutch roll mode satisfies flying quality level 1 regardless of the control methods in all analysis conditions. Likewise, the roll mode time constant is

1.0 s or less, and the spiral root in which spiral mode does not exist is not applicable. The flying qualities satisfy level 1.

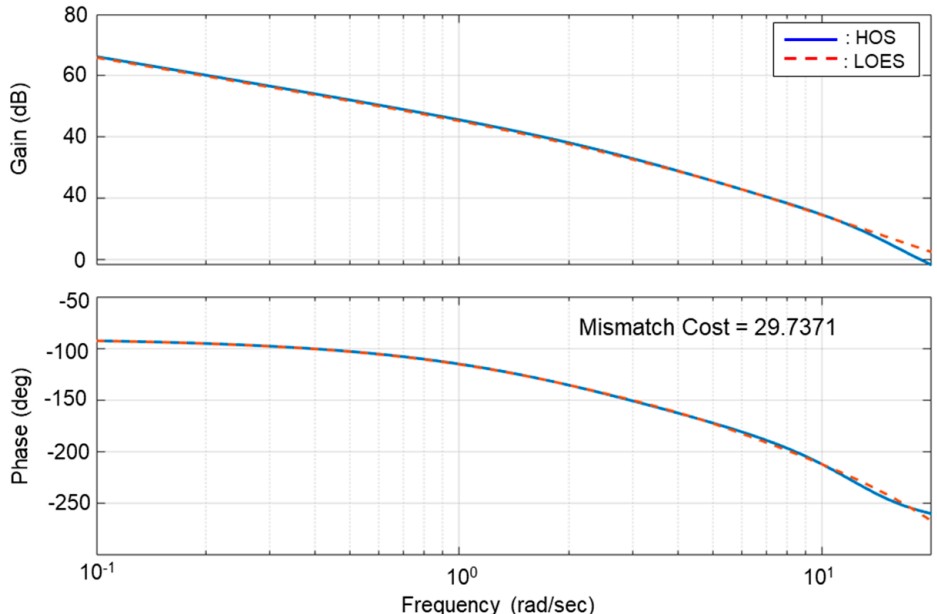

**Figure 6.** Gain and phase plot—HOS vs. LOES (M0.8, 30 K, UA, 1 g, $K_{aug}$ = 1.0).

The bandwidth criterion [11] is a handling quality criterion at the high-frequency band in which the phase margin is at least 45° and the gain margin is at least 6 dB, also providing the criteria for time delay resulting from phase delay in the frequency domain. Figure 7 shows the analysis results of bandwidth for roll and sideslip in flight conditions for each control method. For all the control methods, the change of flying qualities in bandwidth for both roll and sideslip is small and negligible, and the time delay slightly increases as $K_{aug}$ increases. Overall, the bandwidth is less affected regardless of control method type, satisfying flying quality level 1.

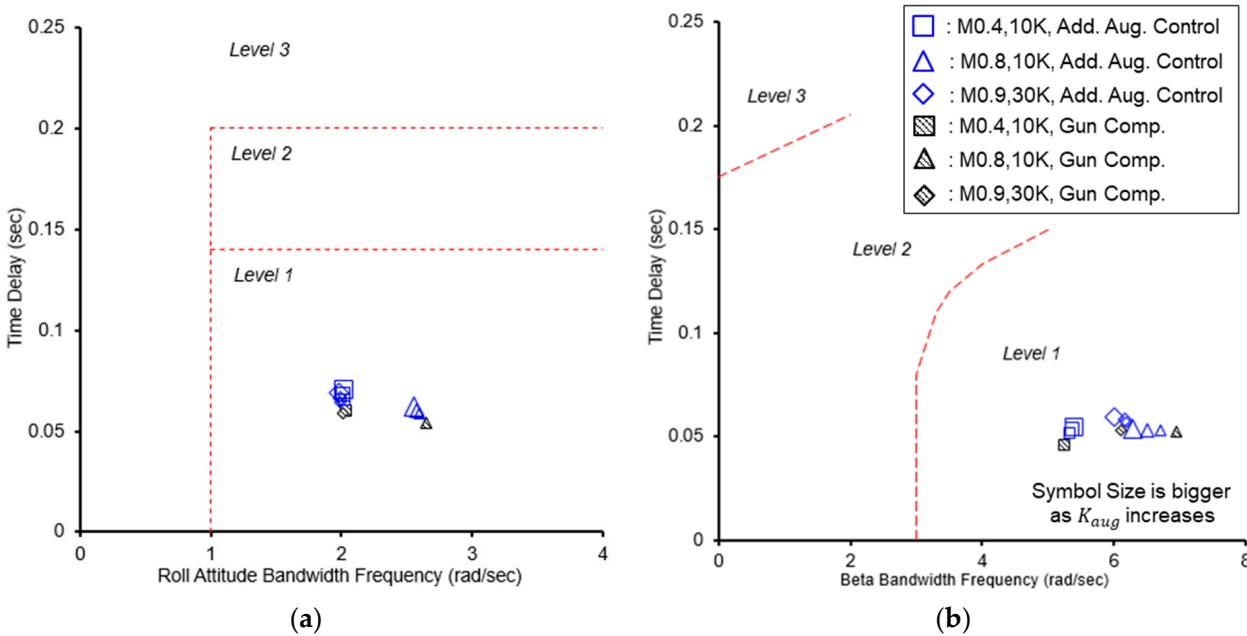

**Figure 7.** Result of bandwidth analysis: (**a**) roll attitude bandwidth; (**b**) sideslip bandwidth.

The Gibson phase rate criterion [41] concerns the open-loop attitude frequency response in the region around the –180° attitude phase. The average phase rate is derived from the excess phase lag between the pilot-induced oscillation (PIO) frequency and twice that frequency. Figure 8 shows the result of the Gibson phase rate in flight conditions for each control method. The Gibson phase rate increases at −180° phase delay frequency as $K_{aug}$ increases, satisfying flying quality level 1 below 50°/Hz.

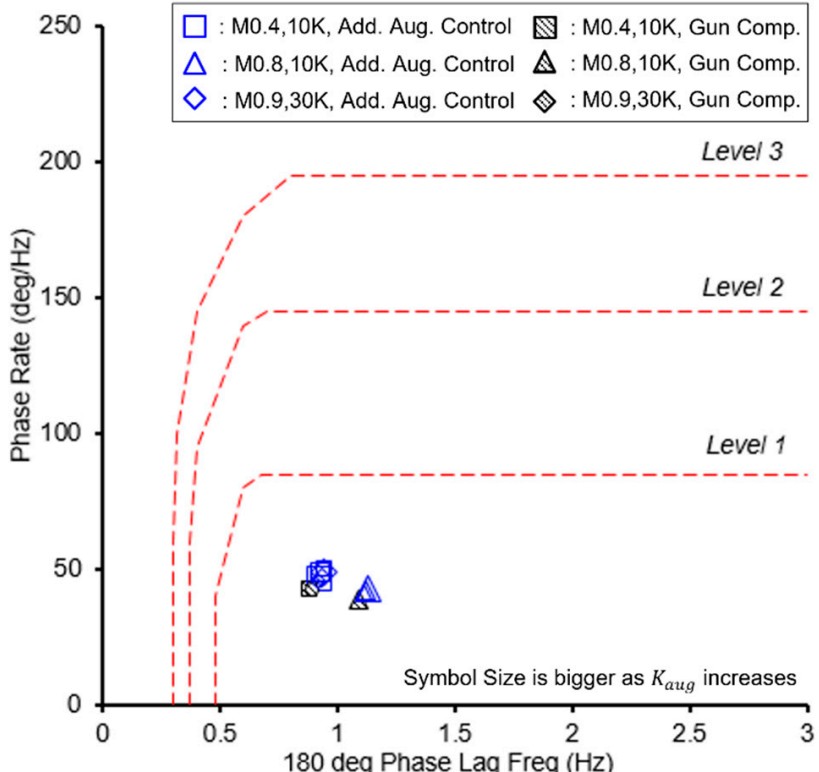

**Figure 8.** Result of Gibson phase rate analysis.

### 4.4. Transient Response Evaluation during Gun Firing

This section presents the results of evaluating the transient response during gun firing for each control method while performing 1 g level flight and slow down turn (SDT) in a flight condition of Mach number 0.8 and altitude of 10 kft. The transient response, the attitude stabilization, and the deviation in the coordinates of aircraft position due to gun firing are analyzed.

Figure 9 shows the simulation results of comparing the transient responses of the aircraft for each control method when the gun is fired for 2 s at Mach number 0.8, altitude 10 kft, UA, and 1 g level flight condition. Here, the black solid line represents the result in a basic case in which even open-loop type gun compensation control is not applied, the red dotted line is the result in a case in which the gun compensation control is applied, and the blue dotted line and the solid line are the results of hydraulic INDI control when $K_{aug}$ values of 0.8 and 1.0 are applied, respectively.

The simulation results for each control method can be summarized as follows: First, when the gun compensation control is not applied, a rapid transient response occurs with a maximum roll rate of −9.1°/s and a maximum yaw rate of −2.4°/s around 3.5 s after gun firing. The aircraft has large deviations of −11° roll attitude, −3.1° yaw attitude, and +0.7° sideslip from the initial level flight condition 10 s after the start of the simulation. Second, when the gun compensation control is applied, the transient response is reduced by more than 40% to −4.2°/s roll rate and −1.7°/s yaw rate at the 3.5 s after gun firing. In addition, the transient responses of the roll rate and yaw rate increase to +5.5°/s and

+2.0°/s around 5 s after the gun firing is finished. The aircraft stabilizes within 0.2° roll attitude, −0.45° yaw attitude, and +0.4° sideslip as the transient response decreases 10 s after the start of the simulation. Third, the additional augmentation control effectively decreases the aircraft transient response during gun firing. When $K_{aug}$ is 0.8, the transient responses of the roll rate and the yaw rate significantly decrease to −2.2°/s and −1.3°/s, respectively. The aircraft stabilizes within −2.0° roll attitude, −0.8° yaw attitude, and +0.2° sideslip after the transient response decreases. However, the aircraft attitude is stabilized close to 0° as $K_{aug}$ increases to 1.0 since the magnitude of the transient response decreases proportionally as $K_{aug}$ increases.

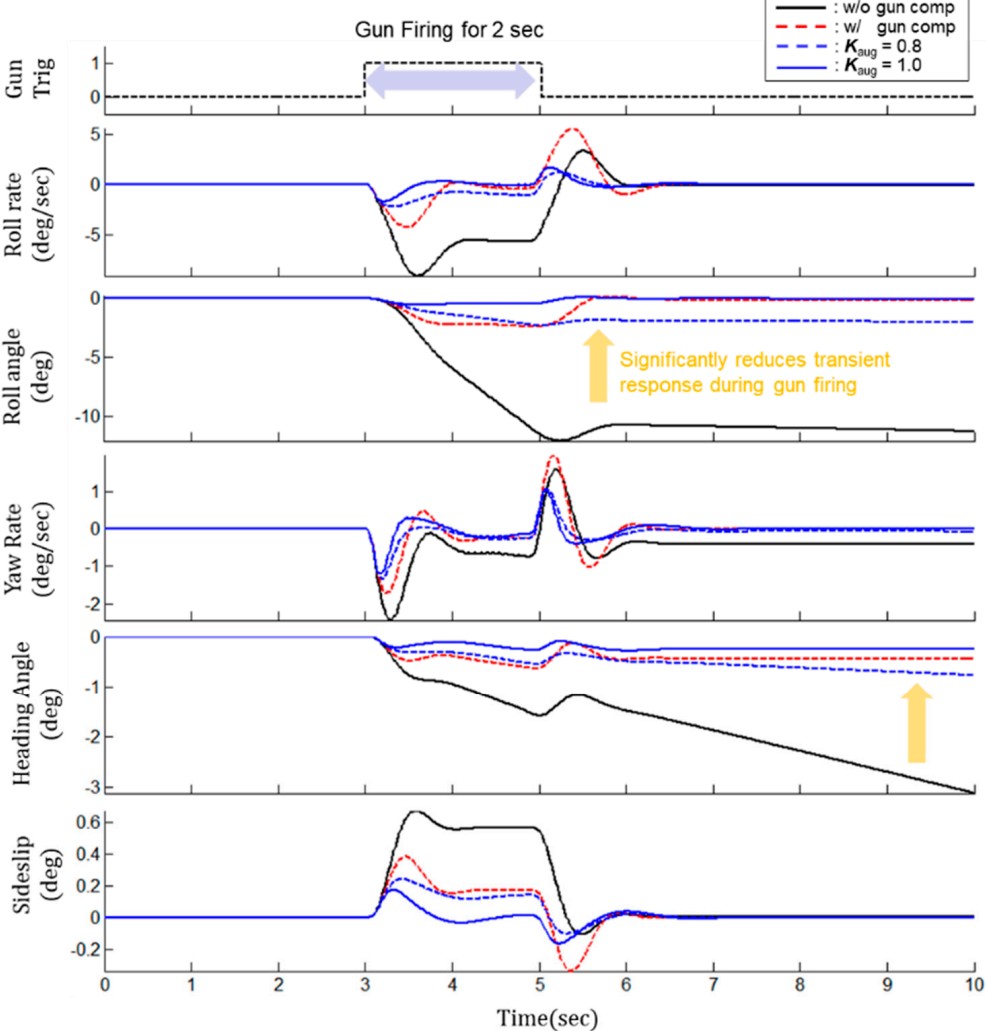

**Figure 9.** Result of lateral-direction transient response during gun firing for each control method at M0.8, 10 kft altitude, 1 g level flight.

Figure 10 shows the simulation results of evaluating the transient responses of the aircraft for each control method when gun firing lasts for 2 s during SDT maneuver at Mach number 0.8, altitude 10 kft, and UA flight condition. Here, the black dotted line represents the result in a case of SDT maneuver without gun firing, the black solid line represents the result in a case of SDT maneuver with gun firing without gun compensation control, the red dotted line represents the result in a case of SDT maneuver with gun firing with the gun compensation control, and the blue dash line and the solid line are the results of the additional augmentation control when $K_{aug}$ values of 0.8 and 1.0 are applied, respectively.

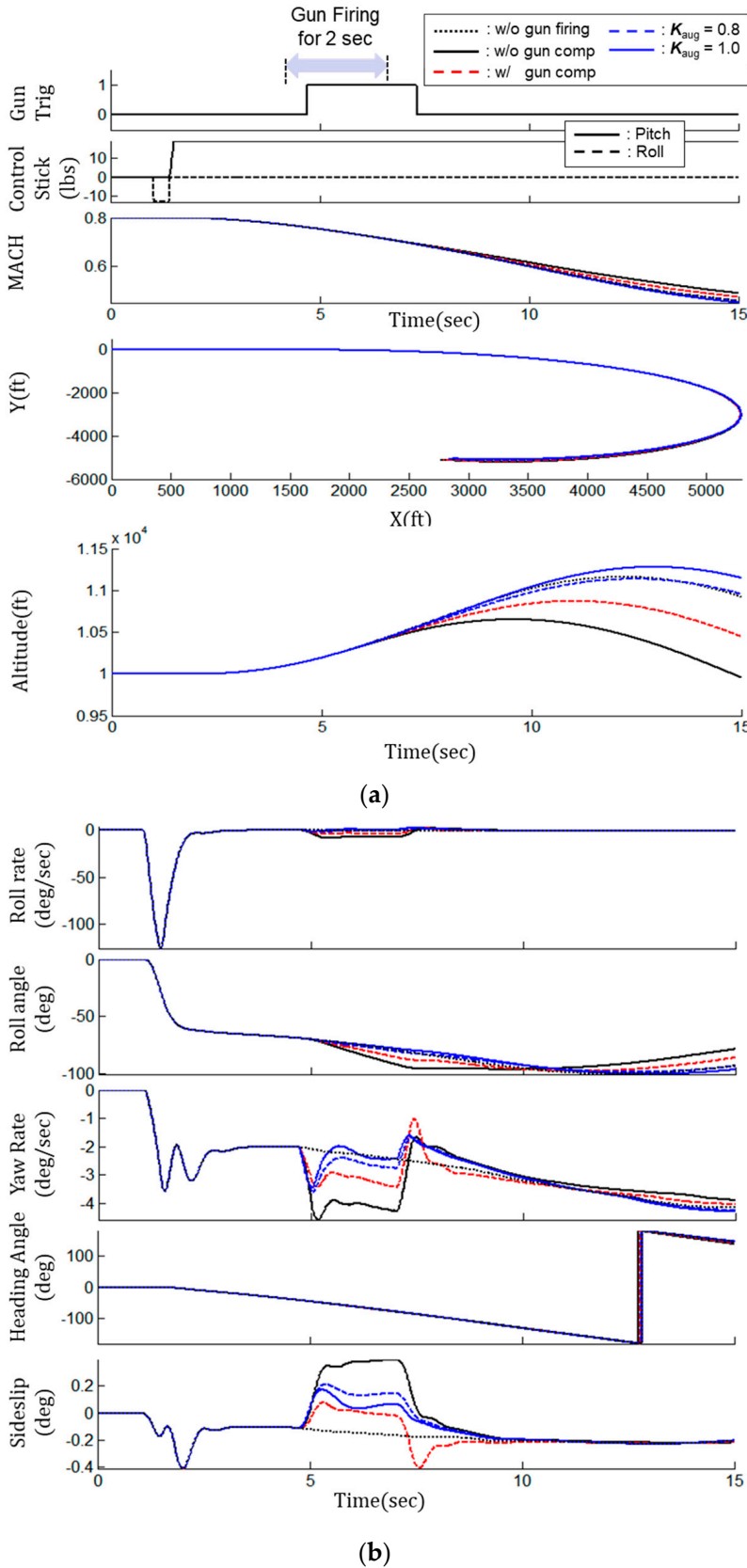

**Figure 10.** Result of simulation during gun firing for slow down turn maneuver at M0.8, 10 kft altitude: (**a**) aircraft trajectory; (**b**) transient response.

The purpose of this evaluation is to analyze how the transient response affects the attitude of the aircraft in the case of gun firing during air-to-air combat maneuvers. The simulation results can be summarized as follows: First, in the case of gun compensation control not being applied, the maximum roll rate and the maximum yaw rate at the time of gun firing are about $-8.8°$/s and $-4.2°$/s, respectively. Due to this transient response, the roll attitude is $-96°$ at 7.5 s; the roll attitude deviates about $-16°$ compared to the case of not firing the gun. The altitude drops to 9950 ft by $-950$ ft at 15 s, compared to the case of not firing the gun. Second, in the application of gun compensation control, the transient responses of the maximum roll rate and maximum yaw rate are reduced to $-4.6°$/s and $-3.4°$/s, respectively. The roll attitude is about $-88°$ at 7.5 s; the roll attitude deviates about $-8°$ compared to the case of not firing the gun. The altitude drop is reduced to 10,450 ft by $-470$ ft at 15 s compared to the case of not firing the gun. Third, in the additional augmentation control with $K_{aug}$ set to 0.8, the roll rate transient response decreases to $-0.5°$/s during gun firing, maintaining the aircraft attitude within $-80°$ roll. The attitude which is 10,920 ft at 15 s is almost similar to the case of no gun firing. The altitude slightly increases up to 11,150 ft when $K_{aug}$ is changed to 1.0, which shows the characteristic that the altitude deviation increases when $K_{aug}$ increases.

As a result of evaluating the transient response during gun firing for each control method, the gun compensation control has the advantage of slightly reducing the transient response at the time the gun firing starts but has also the disadvantage of slightly increasing the transient response at the time the gun firing finishes. On the other hand, the additional augmentation control is significantly effective in reducing the transient responses during gun firing and maintaining the aircraft attitude. In the additional augmentation control, the transient response of the aircraft during gun firing decreases proportionally as $K_{aug}$ increases.

*4.5. Robustness Analysis for Various Uncertainties*

This section presents the results of evaluating the robustness of each control method regarding the transient response characteristics of the aircraft for various uncertainties during gun firing; stability margins required for the control system to allow the various uncertainties in system dynamics are discussed in our previous papers [22,41].

The robustness for each control method is evaluated in uncertainties of the following three aspects: (1) a case of one fixed control gain in flight conditions, (2) a case of reaction force uncertainty which was applied as 25%, and (3) a case of time delay between the gun trigger activation and the actual gun firing. The robustness of the control system against uncertainty is evaluated by quantifying the deviation in aircraft response for each control method at the point where the angular velocity and attitude differences are the highest.

First, Figure 11 is the simulation result of evaluating the robustness of the gun compensation control with one fixed control gain and that of the additional augmentation control with $K_{aug}$ set to 0.8, by varying the Mach number and altitude. Figure 11a shows the results of evaluating the robustness of aircraft on gun firing when one fixed control gain of the gun compensation control designed at Mach number 0.8 and altitude 10 kft is applied to other flight conditions such as Mach number 0.4 and attitude 10 kft, and Mach number 0.9 and attitude 40 kft. The gun compensation control using the fixed control gain without scheduling the optimum control gain according to flight conditions shows a significant difference in the transient response between the flight conditions. At this time, the deviations in the roll rate and yaw rate response at the flight conditions are $4.2°$/s and $1.5°$/s, respectively. The response deviation at 10 s is $13°$ for the roll attitude and $7°$ for the yaw attitude. Figure 11b is the result of evaluating the transient response during gun firing based on the additional augmentation control with $K_{aug}$ set to 0.8, and it shows a significantly robust characteristic against changes in flight conditions. The deviation of roll rate transient response between flight conditions is less than $0.9°$/s, which is reduced by more than 80% compared to the roll rate that occurs when the gun compensation control with the fixed control gain is applied. The deviation of roll attitude is also significantly

reduced to within 1.0°, the deviation of maximum yaw rate is reduced to 0.86°/s, and the yaw attitude is significantly reduced to within 1.4°. Therefore, in the additional augmentation control, the deviation of roll rate can be reduced by more than 50% compared to the gun compensation control.

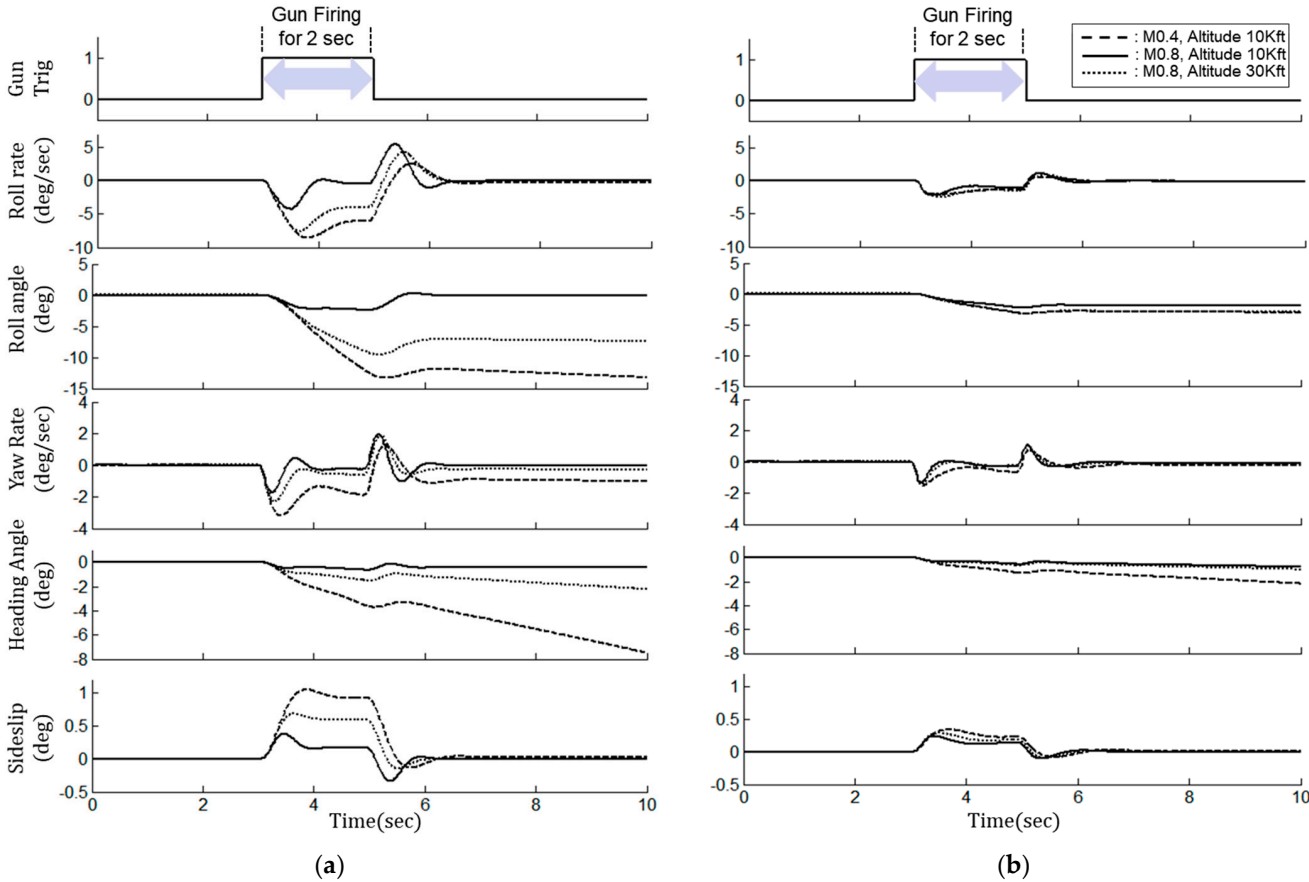

**Figure 11.** Result of robustness in gain scheduling for each control method at M0.4, 10 kft altitude: (**a**) the fixed control gain of the gun compensation control designed at Mach number 0.8 and altitude 10 kft; (**b**) the additional augmentation control with $K_{aug}$ set to 0.8.

Second, Figure 12 is the simulation result of evaluating the robustness of each control method against the uncertainty of the reaction force that may occur during gun firing at Mach number 0.8, attitude 30 kft, and 1 g level flight condition. The uncertainties of reaction force are applied as 25%, in accordance with the airworthiness criteria of MIL-HDBK-516B [42]. Figure 12a shows the result of gun compensation control, and Figure 12b shows the simulation result of the additional augmentation control with $K_{aug}$ set to 0.8. When the reaction force uncertainty occurs during gun firing, the gun compensation control cannot adequately reduce the transient response, so the deviation of transient response is quite large compared to the case where it does not occur. At this time, the deviation of transient response at 4.7 s is 5.3°/s for the roll rate and 1.4°/s for the yaw rate. The deviation of the aircraft attitude is 8° in the roll attitude and 2° in the yaw attitude. On the other hand, in the additional augmentation control as shown in Figure 12b, the deviation of the transient response due to reaction force uncertainty is significantly reduced compared to the gun compensation control to 1.3°/s for roll rate and 0.6°/s for yaw rate. Therefore, in the additional augmentation control, the deviation of the aircraft attitude can be reduced by more than 50% within 1.2° for roll attitude and 0.5° for yaw attitude.

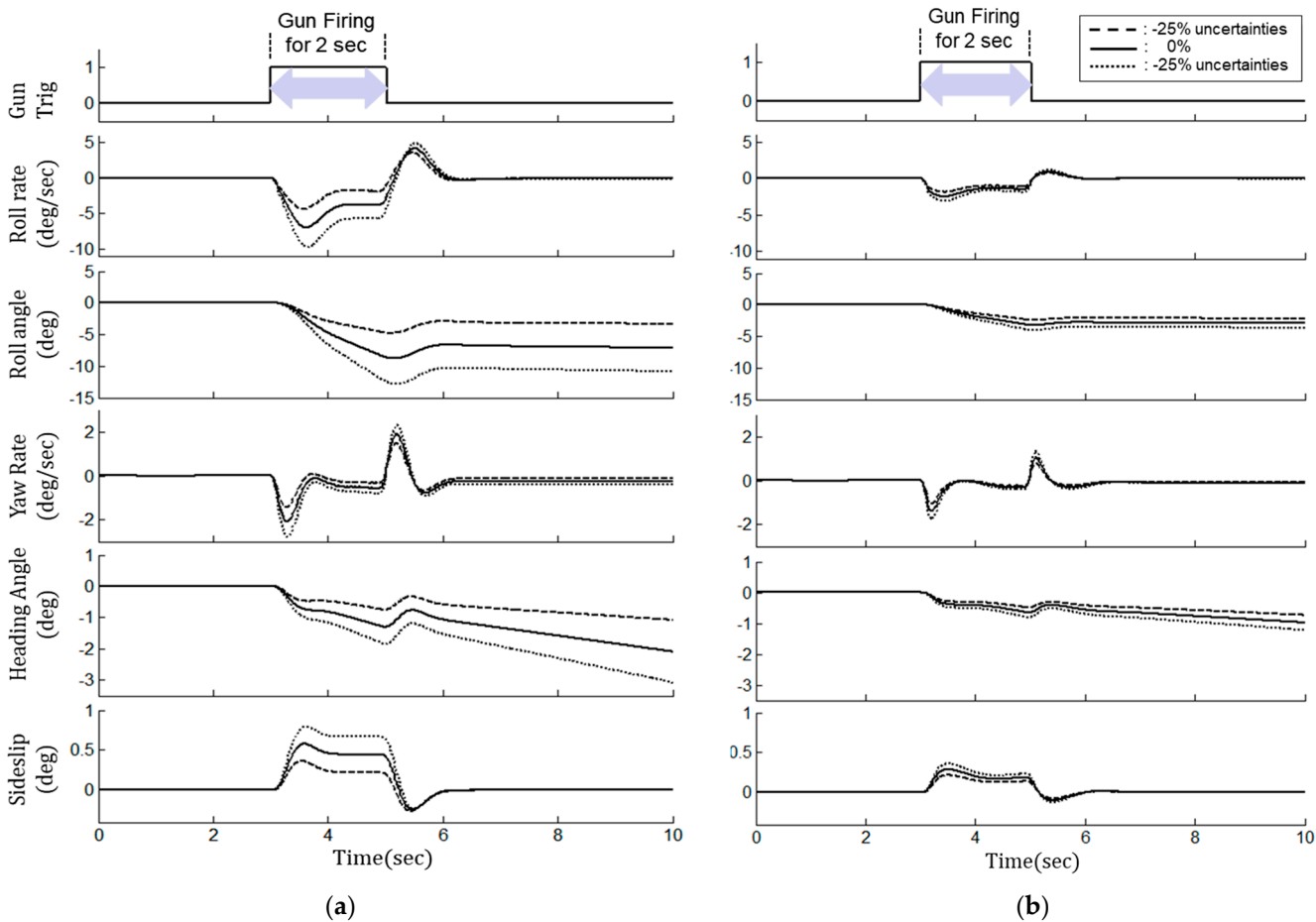

**Figure 12.** Result of robustness in gun firing moment uncertainties for each control method at M0.9, 30 kft altitude: (**a**) gun compensation control; (**b**) additional augmentation control.

Third, Figure 13 shows the simulation results of the effect of time delay for each control method at M0.8, altitude 10 kft, and 1 g level flight condition. There is a time delay between when the pilot activates the gun trigger and when the bullet is actually fired and a reaction force acts on the aircraft. Generally, the amount of time delay is measured from the gun firing test in the flight test and reflected in the gun compensation control law design. The amount of time delay uncertainty to be evaluated is selected as 0 frames, 4 frames (0.0625 ms), and 8 frames (0.0125 ms). Figure 13a shows the evaluation result of the gun compensation control, and Figure 13b shows the evaluation result of additional augmentation control with $K_{aug}$ set to 0.8. The gun compensation control presents that the transient response increases as the time delay increases. The time delay has an adverse effect on the transient response, mainly at the beginning of the gun firing, while it does not at the end of the gun firing. In the transient response, the deviation of transient response is 1.3°/s for the roll rate and 0.3°/s for the yaw rate. The deviation of aircraft attitude is within 0.7° for roll attitude and 0.19° for yaw attitude. On the other hand, the additional augmentation control has little effect on time delay, even though the time delay increases.

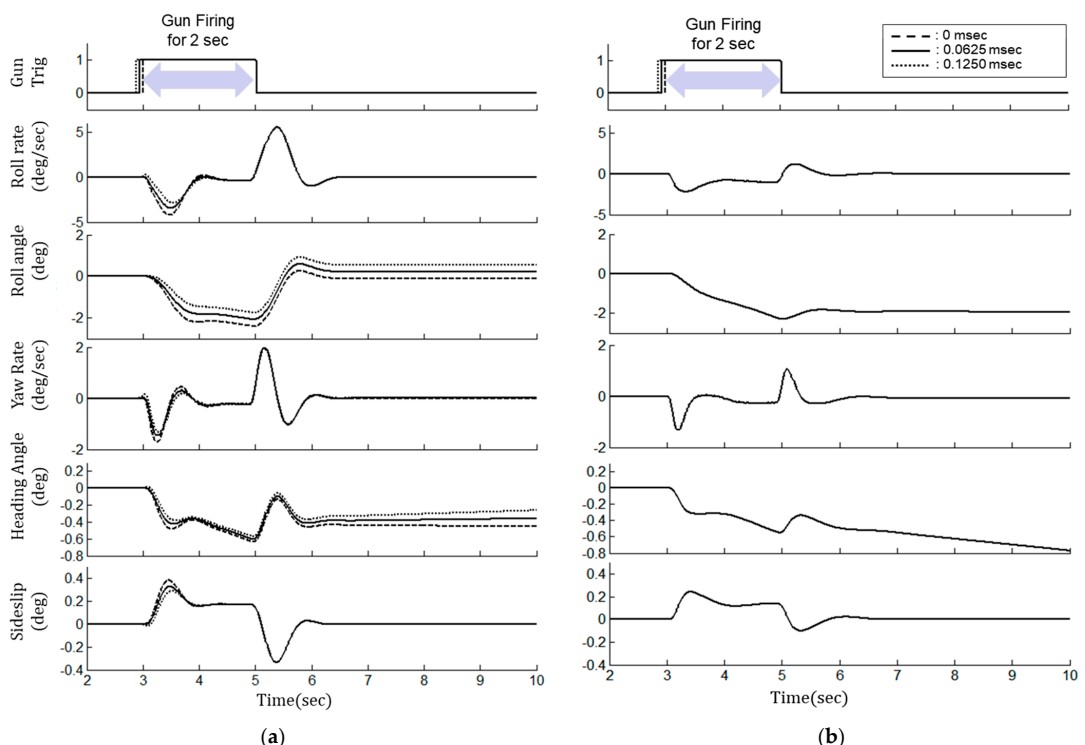

**Figure 13.** Result of robustness in time delay for each control method at M0.8, 10 kft altitude: (**a**) gun compensation control; (**b**) additional augmentation control.

## 5. Conclusions

In the modern highly maneuverable fighter aircraft, guns as well as air-to-air missiles are armed to gain strategic superiority over other fighters in air-to-air combat. A fighter's gun system is used to shoot down targets at close range in dogfighting. However, in a short moment during gun firing, the aircraft has an abrupt transient response for rolling and yawing motions due to reaction force caused by the explosive force since the muzzle of the gun is mounted at an offset from the longitudinal axis of the aircraft. The handling qualities and mission effectiveness are also degraded if the transition response is excessive.

A fighter normally uses open-loop type gun compensation control to reduce the transient response for gun firing. However, in this gun compensation control, the control gain must be scheduled for each flight condition within the operational flight envelope. Otherwise, it is highly affected by the uncertainty of external force caused by gun firing. This is the reason why so many flight tests are required.

In this study, we proposed the application of an additional augmentation control method that combines the model-based INDI control and sensor-based INDI based on the angular acceleration measured from the IMU sensor to minimize the maximum overshoot of the transient response during gun firing. As a significantly robust method against model uncertainties, this additional augmentation control quite effectively reduces the transient response and stabilizes the attitude even when the model of the external force exerted on the aircraft during gun firing is uncertain. In addition, this additional augmentation control has a simple control structure, so it does not need many flight test sorties during aircraft development, unlike the open-loop type gun compensation, since the control gain does not need to be inherently designed according to the gun firing external forces changed according to flight conditions. Therefore, the aircraft development cost and period can be reduced without requiring additional flight tests for complex control gain scheduling of the gun compensation control.

In the future, it is planned to further improve gun targeting accuracy by developing an algorithm for automatic aiming during machine gun firing by integrating weapon control and flight control.

**Author Contributions:** Conceptualization, C.-h.J., C.K. and B.S.K.; methodology, C.-h.J. and C.K.; software, C.-h.J. and C.K.; validation, C.-h.J. and C.K.; formal analysis, C.-h.J. and C.K.; investigation, C.-h.J. and C.K.; resources, C.-h.J. and C.K.; data curation, C.-h.J. and C.K.; writing—original draft, C.-h.J. and C.K.; writing—review and editing, C.-h.J. and C.K.; visualization, C.-h.J. and C.K.; super-vision, B.S.K.; This research received no external funding. All authors have read and agreed to the published version of the manuscript.

**Funding:** This research received no external funding.

**Data Availability Statement:** Data sharing not applicable. No new data were created or analyzed in this study. Data sharing is not applicable to this article.

**Acknowledgments:** The authors would like to deliver their sincere thanks to the editors and anony-mous reviewers.

**Conflicts of Interest:** The authors declare no conflict of interest.

## Nomenclature

| | |
|---|---|
| $\boldsymbol{x}$ | state vector |
| $\boldsymbol{u}$ | control input vector |
| $\boldsymbol{f}$ | nonlinear state dynamic function |
| $\boldsymbol{g}$ | nonlinear control distribution function |
| $\Delta\boldsymbol{u}$ | incremental control command (°) |
| $\Delta\boldsymbol{d}$ | virtual control command (°) |
| $\boldsymbol{u_0}$ | previous control command (°) |
| $\boldsymbol{K}_{aug}$ | additional augmentation control gains |
| $\boldsymbol{f}_{obm}$ | nonlinear state dynamic function of OBM |
| $\boldsymbol{g}_{obm}$ | nonlinear control distribution function of OBM |
| $\dot{\boldsymbol{x}}_{des}$ | rate of desired state vector (°/s$^2$) |
| $\dot{\boldsymbol{x}}_{obm}$ | rate of state vector calculated from OBM (°/s$^2$) |
| $\dot{\boldsymbol{x}}_{add}$ | rate of state vector of additional augmentation control sensor (°/s$^2$) |
| $K_{pfn}$ | pilot prefilter numerator gain |
| $K_{pfd}$ | pilot prefilter denominator gain |
| $K_f$ | forward gain |
| $K_{ni}$ | integral gain to normal acceleration |
| $K_{np}$ | proportional gain to normal acceleration feedback |
| $K_q$ | proportional gain to pitch rate feedback |
| $K_{r1}$ | flying quality parameter of roll command |
| $K_{r2}$ | flying quality parameter of roll rate feedback |
| $K_{y1}$ | flying quality parameter of yaw command |
| $K_{y2}$ | flying quality parameter of sideslip feedback |
| $K_{y3}$ | flying quality parameter of sideslip rate feedback |
| $p_s$ | stability axis roll rate (°/s) |
| $p_{s,cmd}$ | stability axis roll rate command (°/s) |
| $\tau_{roll}$ | roll time constant (s) |
| $\beta$ | angle of sideslip (°) |
| $\beta_{cmd}$ | angle of sideslip command (°) |
| $\omega_{dr}$ | Dutch roll frequency (rad) |
| $\zeta_{dr}$ | Dutch roll damping ratio |
| $\dot{q}$ | pitch angular acceleration (°/s$^2$) |
| $\dot{p}$ | roll angular acceleration (°/s$^2$) |
| $\dot{r}$ | yaw angular acceleration (°/s$^2$) |
| $\dot{q}_{des}$ | desired pitch angular acceleration (°/s$^2$) |
| $\dot{p}_{des}$ | desired roll angular acceleration (°/s$^2$) |
| $\dot{r}_{des}$ | desired yaw angular acceleration (°/s$^2$) |
| $p$ | roll rate (°/s) |
| $q$ | pitch rate (°/s) |
| $r$ | yaw rate (°/s) |

| | |
|---|---|
| $I_{ii}$ | principal moment of inertia (slug-ft$^2$) ($i = x, y, z$) |
| $I_{ij}$ | production moment of inertia (slug-ft$^2$) ($i = x, y, z, j = x, y, z$) |
| $L$ | rolling moment of the aircraft |
| $N$ | yawing moment of the aircraft |
| $L'_k$ | rolling moment for k ($k = \beta, p, r, \delta_{ea}, \delta_r$) |
| $N'_k$ | yawing moment for k ($k = \beta, p, r, \delta_{ea}, \delta_r$) |
| $\delta_k$ | control surface deflection for k ($k = ea, aa, r$) |
| $K_{aug}$ | additional augmentation control gains |
| $G^u_{ac}$ | aircraft plant dynamics for control surface |
| $\dot{x}_{fb}$ | total angular acceleration feedback ($°/s^2$) |
| $H_{sync}$ | synchronization filter matrix |
| $\zeta_{syn}$ | damping ratio of 2nd order synchronization filter |
| $\omega_{syn}$ | natural frequency of 2nd order synchronization filter (rad) |
| $T_{\theta2}$ | pitch attitude time constant |
| $T^{des}_{\theta2}$ | desired pitch attitude time constant |
| $\Delta L_{gun}$ | resultant additional rolling moment from gun burst |
| $\Delta M_{gun}$ | resultant additional pitching moment from gun burst |
| $\Delta N_{gun}$ | resultant additional yawing moment from gun burst |
| $\delta\Delta_{e, gun}$ | additional required symmetric horizontal control surface deflections |
| $\Delta\delta_{ea, gun}$ | additional required asymmetric horizontal control surface deflections |
| $\Delta\delta_{aa, gun}$ | additional required aileron control surface deflections |
| $\Delta\delta_{r, gun}$ | additional required rudder control surface deflections |
| $K_{ea}$ | control gain ratio of asymmetric horizontal control surface deflections |
| $sig_{GT}$ | gun burst signal |
| $\tau_g$ | time lag from gun burst signal to actual firing of the gun |
| $K_{gx}$ | control gains of gun compensation control |

**List of Acronyms**

| | |
|---|---|
| AMRAAM | advanced medium-range air-to-air missile |
| HUD | head-up display |
| HQ | handling quality |
| INDI | incremental nonlinear dynamic inversion |
| IMU | inertial measurement unit |
| OML | outer mold line |
| AoA | angle of attack |
| AoS | angle-of-sideslip |
| CA | control allocation |
| RLSN | recursive linear smoothed Newton |
| DLR | German Aerospace Center |
| HARV | high angle-of-attack research |
| HOS | high-order system |
| LOES | low-order equivalent system |
| FLCC | flight control computer |
| JSF | joint strike fighter |
| LOES | low-order equivalent system |
| N/A | not applicable |
| NASA | National Aeronautics and Space Administration |
| NLR | Netherlands Aerospace Centre |
| OBM | onboard model |
| RESTORE | reconfigurable control for tailless aircraft |
| RSRI | rolling surface-to-rudder interconnect |
| SDT | slow down turn |
| STOVL | short take-off/vertical landing |
| VAAC | vectored thrust aircraft advanced control |

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
