# Peer review of "Flight Control Law for Stabilizing Transient Response of the Aircraft during Gun Firing"

_aerospace, doi:10.3390/aerospace10040365_

Round 1
Reviewer 1 Report
The paper could be more synthetic, there are several repetitions. The link between flying qualities and compensation is not always clear.
line 49-55 : Does it concern gun or missile ?
134 : NDI or INDI ?
211 : change word "form" in "from"
220 : the notation g^-1 usually refers to the inverse function, here it's (g(x))^-1
228 : needs more explanation why u_0 = 0 ?
253 : x = x_0
280 : The equation 14 doesn't match with the previous figure.
292 - 296 : paragraph should be move in the next part
§ 3.3.2 : The link between "Desired dynamics" and gun compensation is not clearly explained.
372 : (g) is not an unit.
379 : two times sideslip
403 - 406 : sentence not correct
434 : modeling of DeltaN_gun and DeltaL_gun are not explained, there is just figure with oscillations ?
453 : control gain optimisation is not explained
478 : the table 2 give N/A for the spiral mode : why ? you have a spiral root (eigenvalue ?) of 0.035
Figure 6 : you mention M=0.9 ?
Figure 6 represents gain and phase of which function ?
485 - 501 : not clear
Figure 7b : legend is two times Add. Aug Control
On figure 10 : what is the damping of the Dutch Roll ? It's seem to to be les than 0.7
614 : how uncertainty is modelize ?
Author Response
We appreciate your reviews on our paper. The comments provided were very useful in revising our manuscript so that we can improve the shortcomings of the manuscript. We reflected all of your comments in our paper.
Thank you so much for your comments.

Reviewer 2 Report
The paper is very interesting . The authors made good effort in writing the paper and producing the figure. However, I have some comments such as:
1- in figure 1 please add the label for x axis.
2- nothing was mentioned about how to obtian the gun firing addational moments and how to calculate them.
3- I think the lines from 517 to 533 would be better if summarized in a table.
4- the same for lines 546 to 562 .
5- the English needs polishing.
Author Response

(The authors gave the same response as above.)

Reviewer 3 Report
This paper is notable and interesting in the scope of this Journal. However, it has major shortcomings. In that regard, the following points must be clarified/corrected:
1- The introduction should be rewritten, and the novelty of the research should be clearly mentioned in the last second paragraph.
2-Are equations (1-3) valid for transient dynamics, or are they valid for a steady condition? The time of firing is not usually large; therefore, it is better to see its effect on transient dynamics.
3-The performance of the angular estimation system in the presence of uncertainties and sensor noise should be investigated.
4- It is not clear what the effect of the sensor noise is on the outputs and there is not any information about the noise.
5- There is no comparison between other similar methods, such as the method introduced in "Modified adaptive discrete-time incremental nonlinear dynamic inversion control for quad-rotors in the presence of motor faults."
6- At the end of page 5, "I consider the method of differentiating the angular velocity from the Inertial Measurement Unit sensor to obtain the angular acceleration." There is not any equation for that; is it obtained just by differentiating methods? If so, how do we deal with the effect of amplifying the noise?
7- In figure 2, there are 2 blocks (control effectiveness estimation, angular estimation, and control allocation), but there are not any clear equations related to these blocks.
8- There is not any stability proof or stability analysis.
Author Response

(The authors gave the same response as above.)

Round 2
Reviewer 3 Report
I appreciate the authors' responses to each comment.
The manuscript, in my opinion, is prepared for publishing in this journal.